# General practitioners' perceptions of using virtual primary care during the COVID-19 pandemic: An international cross-sectional survey study

Edmond Li[1]*, Rosy Tsopra[2,3,4], Geronimo Jimenez[5,6], Alice Serafini[7], Gustavo Gusso[8], Heidrun Lingner[9], Maria Jose Fernandez[10,11,12], Greg Irving[13], Davorina Petek[14], Robert Hoffman[15], Vanja Lazic[16], Ensieh Memarian[17], Tuomas Koskela[18], Claire Collins[19], Sandra Milena Espitia[20], Ana Clavería[11,12,21], Katarzyna Nessler[22], Braden Gregory O'Neill[23], Kyle Hoedebecke[24], Mehmet Ungan[25], Liliana Laranjo[26,27], Saira Ghafur[1], Gianluca Fontana[1], Azeem Majeed[28], Josip Car[5,10,11,12], Ara Darzi[1], Ana Luisa Neves[1]*

1 Institute of Global Health Innovation, Faculty of Medicine, Imperial College London, London, United Kingdom, 2 INSERM, Université de Paris, Sorbonne Université, Centre de Recherche des Cordeliers, Information Sciences to support Personalized Medicine, F-75006 Paris, France, 3 Inria Paris, Paris, France, 4 Department of Medical Informatics, Hôpital Européen Georges-Pompidou, AP-HP, Paris, France, 5 Center for Population Health Sciences (CePHaS), Lee Kong Chian School of Medicine, Nanyang Technological University Singapore, Singapore, 6 Department of Public Health and Primary Care, Leiden University Medical Center, Leiden, Netherlands, 7 Local Health Authority of Modena, Italy, 8 Universidade de Sao Paulo, Sao Paulo, Brazil, 9 Hannover Medical School, Center for Public, Health and Healthcare, German Center for Lung Research (DZL) / BREATH Hannover, Germany, 10 Leiro Health Center, Leiro, Spain, 11 Galicia South Health Research Institute, Vigo, Spain, 12 Primary Care Prevention and Health Promotion Network (redIAPP), Spain, 13 Health Research Institute, Edge Hill University, Ormskirk, United Kingdom, 14 Department of Family Medicine, Faculty of Medicine, University of Ljubljana, Ljubljana, Slovenia, 15 Department of Family Medicine, Sackler Faculty of Medicine, Tel Aviv University, Tel Aviv, Israel, 16 University of Zagreb, Zagreb, Croatia, 17 Department of Clinical Sciences in Malmö, Lund University, Internal Medicine- Epidemiology Research Group, Skane University Hospital, Malmö, Sweden, 18 General Practice, Tampere University, Faculty of Medicine and Health Technology and Tampere University Hospital, Finland, 19 Irish College of General Practitioners, Dublin, Ireland, 20 Colombian Family Medicine Society, Colombia, 21 Primary Care Research Unit. Vigo Health Area, Vigo, Spain, 22 Department of Family Medicine, Jagiellonian University Medical College, Krakow, Poland, 23 MAP Centre for Urban Health Solutions, Li Ka Shing Knowledge Institute, St. Michael's Hospital, Toronto, Canada, 24 Department of Utilization Management, Oscar Health, Dallas, United States of America, 25 Department of Family Medicine, Ankara University School of Medicine, Ankara, Turkey, 26 Westmead Applied Research Centre, Faculty of Medicine and Health, The University of Sydney, Sydney, Australia, 27 Australian Institute of Health Innovation, Macquarie University, Sydney, Australia, 28 Department of Primary Care & Public Health, School of Public Health, Imperial College London, London, United Kingdom

* edmond.li19@imperial.ac.uk (EL); ana.luisa.neves14@ic.ac.uk (ALN)

**Data Availability Statement:** Data is available upon reasonable request. The datasets generated and/or analysed during the current study will be

## Abstract

With the onset of COVID-19, general practitioners (GPs) and patients worldwide swiftly transitioned from face-to-face to digital remote consultations. There is a need to evaluate how this global shift has impacted patient care, healthcare providers, patient and carer experience, and health systems. We explored GPs' perspectives on the main benefits and challenges of using digital virtual care. GPs across 20 countries completed an online questionnaire between June–September 2020. GPs' perceptions of main barriers and challenges were explored using free-text questions. Thematic analysis was used to analyse the data. A total of 1,605

made available upon reasonable request through the Imperial College London Big Data Analytical Unit (BDAU) data depository.

**Funding:** This work was supported by the National Institute for Health Research (NIHR) Patient Safety Translational Research Centre (PSTRC) (PSTRC-2016-004) and a Research Grant from the European General Practice Research Network (EGPRN) (2020/19), and infrastructure support from the NIHR Biomedical Research Centre. The funders/sponsors had no role in the development and drafting of this manuscript.

**Competing interests:** The authors have declared that no competing interests exist.

respondents participated in our survey. The benefits identified included reducing COVID-19 transmission risks, guaranteeing access and continuity of care, improved efficiency, faster access to care, improved convenience and communication with patients, greater work flexibility for providers, and hastening the digital transformation of primary care and accompanying legal frameworks. Main challenges included patients' preference for face-to-face consultations, digital exclusion, lack of physical examinations, clinical uncertainty, delays in diagnosis and treatment, overuse and misuse of digital virtual care, and unsuitability for certain types of consultations. Other challenges include the lack of formal guidance, higher workloads, remuneration issues, organisational culture, technical difficulties, implementation and financial issues, and regulatory weaknesses. At the frontline of care delivery, GPs can provide important insights on what worked well, why, and how during the pandemic. Lessons learned can be used to inform the adoption of improved virtual care solutions and support the long-term development of platforms that are more technologically robust and secure.

## Author summary

Whether it be a simple telephone call or more sophisticated video conferencing systems, virtual care tools have been in use in primary care settings worldwide in one form or another throughout the past two decades. Over time, these tools have grown in availability, matured in their capabilities, but played a largely supportive role as an alternative option to traditional face-to-face consultations. This all changed in early 2020.

The onset of COVID-19 presented a unique opportunity globally which put virtual care tools at the forefront of primary care delivery. The need for social distancing to limit disease transmission resulted in virtual care tools becoming the primary means with which to continue providing primary care services. Hence, our study's goal was to capture the spectrum of GP experiences using virtual care tools during the initial months of the pandemic so as to better understand the perceived benefits and challenges, and explore what changes are needed to allow them to reach their fullest potential.

To this end, we received a total of 1,605 responses from 20 countries globally. Our results demonstrated that virtual care tools were beneficial in limiting COVID-19 transmission, improved convenience when communicating with patients, and encouraged the further adoption of virtual care tools in primary care. Challenges included patients' preferences for face-to-face consultations, digital exclusion of certain populations, diagnostic challenges associated with the inability to perform physical examinations, and their general unsuitability for certain types of consultations. Practical challenges such as higher workloads, payment issues, and technical difficulties were also reported.

Learning from this global natural experiment is critical to both updating existing and introducing new health technology policies concerning virtual primary care. Doing so will be imperative to supporting and promoting the better use of these novel technologies in our evolving healthcare milieu.

## Introduction

### Background

For decades, there have been many initiatives to implement virtual care into healthcare systems. In the US, Kaiser Permanente offers secure email communication and routine telephone

and video consultations [1]. In the UK, the use of telephone consultations is commonplace [2]. Many other healthcare systems worldwide have advocated similarly for a virtual approach [3–9].

The use of virtual care, either via telephone, video, or online technologies, has potential implications on the six domains of quality of care: timeliness, efficiency, patient-centredness, effectiveness, safety, and equity [10]. Virtual consultations can reduce delays in the diagnosis and treatment, thus improving timeliness [11]. They can also facilitate access for patients living in isolated areas, and reduce inequities in care delivery [12–14]. Virtual care can improve primary care efficiency by acting as a gatekeeper by remotely triaging patients, identifying those who require urgent face-to-face care from those who can be managed virtually [15–17]. While some studies have suggested that virtual care can improve efficiency and generate time savings [18,19], others did not find a statistically significant reduction [20]. Virtual care can support the delivery of more patient-centred care and the development of self-management skills [21,22]. Virtual care can be effective in the management of chronic conditions, including chronic obstructive pulmonary disease, heart failure, and diabetes mellitus [23]. Studies have also suggested improvements in patient safety and a potential reduction in hospital admissions [20,23]. However, despite the promised benefits, virtual care has been integrated slowly into primary care. A wide range of obstacles has limited widespread adoption, including cultural, regulatory and policy, industrial and technical, knowledge, financial, and market-related barriers [24].

With the onset of COVID-19, the primary care landscape was radically transformed [25,26]. Many countries have released national guidance encouraging the use of virtual triage and consultation systems [26]. In a few weeks, General Practitioners (GPs) and patients worldwide transitioned from face-to-face consultations to virtual care [27]. Whilst previous evidence surrounding the use of virtual care in general practice came from relatively small and local clinical trials, this pandemic forced patients, healthcare providers, and healthcare systems to embrace virtual consultations as the primary route to access care. Thus, this presents us with a unique opportunity to learn more from this global real-life experiment, identify the main challenges and benefits experienced, and incorporate these lessons into the future of virtual primary care [28].

### Aim

This study aimed to explore GPs' perspectives on the main benefits and challenges of using virtual care tools (i.e., telephone, online consultations tools, messaging platforms etc.,), mapping them against the main domains of quality of care whenever possible. While underutilised in the past due to the dominance of interviewing in qualitative research and misplaced assumptions about lack of data depth, online surveys are now a recognised method for qualitative research [29]. Qualitative surveys typically use open-ended questions to produce long-form answers to capture opinions, experiences, and narratives.

### Methods

The study used an online questionnaire survey of GPs in twenty countries. Recruitment took place between June–September 2020. For further information detailing the rationale and methodology underpinning this study, a protocol paper was recently published [30].

### Study population

The inSIGHT Research Group is a worldwide collaboration of primary care researchers exploring the impact of the COVID-19 pandemic on the adoption of virtual primary care. The

research group is spread across 20 countries (Australia, Brazil, Canada, Chile, Colombia, Croatia, Finland, France, Germany, Ireland, Israel, Italy, Poland, Portugal, Slovenia, Spain, Sweden, Turkey, United Kingdom, United States). Participants were eligible for the survey if they were GPs working in these countries between March and September 2020.

## Sampling

Each local lead sent an email invitation to GPs in their country and shared the link to the survey in social media channels (e.g., LinkedIn, Twitter, Facebook). Local leads who had difficulty achieving the minimum number required (n = 386) used snowballing to increase the number of responses [30]. Snowballing is a recognised technique for recruiting hard-to-reach populations in health studies [31–33].

## Description of questionnaire

The questionnaire included 30 items assessing GPs' perspectives on the adoption and experience of virtual care solutions during the COVID-19 outbreak (**S1 Appendix**). Participants' characteristics were collected, including age, gender, country, practice setting, number of years of experience as GP, and involvement in teaching activities. GPs' perceptions on the main benefits and challenges of using virtual care were assessed using free-text questions ("Which were the main benefits of using virtual care?" / "Which were the main challenges of using virtual care").

## Data analysis

Participants' characteristics were analysed using descriptive statistics. Two independent researchers systematically reviewed the transcripts using the framework analysis method, which includes five main stages: familiarisation, identifying a thematic framework, indexing, charting, and mapping and interpretation [19]. At every stage of the data analysis process, the coding framework was kept deductive and inductive, allowing the inclusion of emergent themes. The coding tree was shared between all researchers for iterative refinement until consensus was reached. Data saturation was reached upon agreement amongst the researchers that no further novel themes emerged toward the end of their independent analyses. Resultant themes, subthemes, and the relationships between them will be visualised using the Miro online whiteboard application [34]. As participants did not provide consent for further contact, it was not possible to ask them to provide feedback on the findings. The Consolidated Criteria for Reporting Qualitative Studies checklist was used to ensure the study meets the recommended standards of qualitative data reporting.

## Ethics

Overall ethical approval for this project was granted by the Imperial College Research Ethics Committee (ICREC) (Reference 20IC5956). This is a dedicated ethics oversight body at Imperial College London for all health-related research involving human participants. In addition, whenever necessary, local ethical approval and relevant permissions in the respective participating countries were also obtained.

Written informed consent was obtained from the participants via an online electronic form placed before the survey itself. Upon consenting, participants were provided access to the survey for two weeks. No minors were involved in this study.

# Results

## Participants' characteristics

1,605 participants participated in the questionnaire. Most respondents (79.3%) were aged 30–59 years, and 60.9% were female (n = 978). Most of the participants have been working as GPs for a minimum of 5 years (79.1%, n = 1,329) and reported being involved in teaching activities (63.7%, n = 1,023). More than half (62.5%) worked in an urban setting (n = 1,004). A full description of the participants, including a breakdown per country, is shown in **Table 1**.

## Main benefits

Benefits clustered around three main themes (**Fig 1**, **Textbox 1**): benefits for quality and safety of care (i.e., safety, effectiveness, equity, efficiency, timeliness, patient-centredness), for health care professionals, and for health care systems.

**Benefits for quality of care.**   The reduced risk of COVID-19 transmission was identified as the main safety benefit. Participants also recognised that virtual care had benefits on effectiveness, ensuring accessibility and continuity of care for both COVID-19 and non-COVID-19 patients. Participants highlighted that virtual care has improved equity in access to care for some groups of patients (e.g., frail elderly people, those with mobility issues, or living far from clinics or in geographically isolated areas). Improvements concerning the efficiency of care included the ability to perform remote triage, reduce unnecessary face-to-face visits (i.e., mild illnesses, prescription renewal, or administrative tasks), and optimise the use of human resources (i.e., enhancing communication between providers). Participants also believed that virtual care improved timeliness, including less time spent in physical dislocation, waiting for administrative procedures, or for clinical appointments. Participants described several benefits for patient-centredness, such as improved convenience and communication, and a positive effect on patient-doctor relationship–often against their prior expectations. GPs acknowledged the importance of virtual care on patient empowerment by increasing self-care awareness for minor illnesses and improving self-management.

**Benefits for healthcare professionals.**   Some respondents identified the flexibility to work remotely in a location of their choosing was a major benefit, as well as having more control over their schedule.

**Benefits for healthcare systems.**   Many respondents underscored the use of virtual care gained during the COVID-19 pandemic has been a major factor hastening the digital transformation (i.e., increasing awareness and trust, improving digital skills of both patients and providers, and upgrading technical capacity). Some participants mention the quick deployment of new digital opportunities (e.g., access to e-referrals, e-prescriptions, and electronic processing of fit-to-work certificates). Other benefits included changes in legal and regulatory frameworks, particularly in what concerns the legal context and remunerations of virtual care tasks.

## Main challenges

Challenges were broadly summarised into three main areas (**Fig 2**, **Textbox 2**): for quality of care, for healthcare providers, and for health care systems.

**Quality of care.**   Participants were concerned that virtual care can negatively impact some aspects of patient-centredness, including patient preferences and the patient-doctor relationship. In fact, one of the identified detractors to the use of virtual care was patients' preference for traditional face-to-face consultations. GPs also reported that it was often difficult to gauge a patient's body language and emotions through digital video/audio channels, thus impairing their ability to build rapport, express empathy, and provide more holistic care to their patients.

**Table 1. Participants' characteristics.**

| Characteristics | n (%) |
|---|---|
| **Gender, n (%)** | |
| Female | 978 (60.9) |
| Male | 616 (38.4) |
| No response | 10 (0.6) |
| Other | 2 (0.1) |
| **Age category, n (%)** | |
| Under 30 | 101 (6.3) |
| 30–39 | 531 (33.1) |
| 40–49 | 415 (25.8) |
| 50–59 | 327 (20.4) |
| 60–69 | 210 (13.1) |
| 70+ | 18 (1.1) |
| Prefer not to answer | 4 (0.2) |
| **Country, n (%)** | |
| Australia | 99 (6.2) |
| Brazil | 53 (3.3) |
| Canada | 53 (3.3) |
| Chile | 58 (3.6) |
| Colombia | 63 (3.9) |
| Croatia | 62 (3.9) |
| Finland | 54 (3.4) |
| France | 62 (3.9) |
| Germany | 50 (3.1) |
| Ireland | 267 (16.6) |
| Israel | 79 (4.9) |
| Italy | 97 (6) |
| Poland | 66 (4.1) |
| Portugal | 95 (5.9) |
| Slovenia | 77 (4.8) |
| Spain | 100 (6.2) |
| Sweden | 76 (4.7) |
| Turkey | 63 (3.9) |
| UK | 77 (4.8) |
| United States of America | 54 (3.4) |
| **Setting, n (%)** | |
| Mixed | 358 (22.3) |
| Rural | 244 (15.2) |
| Urban | 1004 (62.5) |
| **Experience, n (%)** | |
| < 5 years | 336 (20.9) |
| 5–10 years | 359 (22.4) |
| 10–15 years | 242 (15.1) |
| 15–20 years | 174 (10.8) |
| > 20 years | 495 (30.8) |
| **Teaching activities, n (%)** | |
| Yes | 1023 (63.7) |
| No | 569 (35.4) |
| Prefer not to answer | 14 (0.9) |

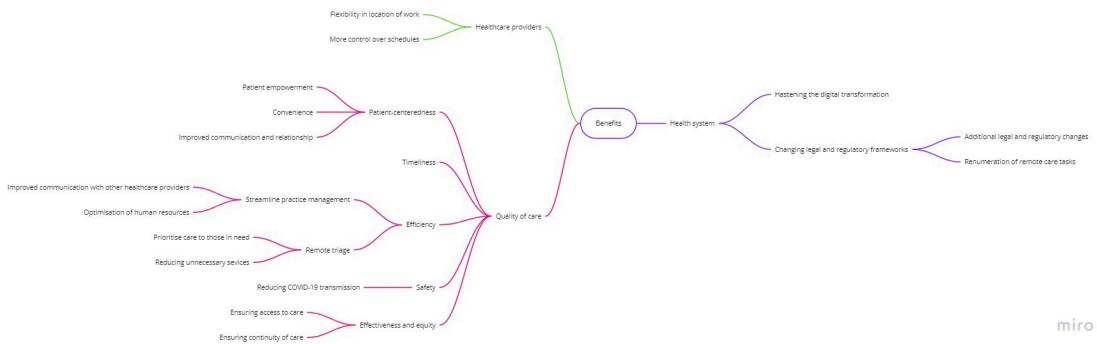

**Fig 1. Conceptual map of main benefits identified.**

GPs acknowledge this to have a potential negative impact on patients' wellbeing since physically attending appointments was an opportunity for social interaction for socially isolated individuals. For clinicians, remote consultations posed additional challenges for patient communication, particularly in emotionally difficult situations, or when there was a need to ensure that their medical advice was properly understood.

Participants expressed concerns regarding a negative impact on equity, particularly regarding the digital exclusion of vulnerable patients. They noted that many patients lack access to reliable hardware (including internet service, computers or smartphones). Even in circumstances where the above-cited hardware is present; some patients may lack the digital literacy and skills needed to independently set up and use relevant digital software.

The most prominent safety challenge was the inability to perform clinical examinations and assess physical signs to inform clinical decision-making. While some of these issues could be mitigated through careful history taking and patient self-reporting (e.g., photographs of visible lesions and remote monitoring devices, such as blood pressure monitors, glucometers, oximeters), GPs mention that few patients had the necessary devices, and if they had, would often struggle to use them and report the results back. Dealing with uncertainty, a recognised challenge in primary care, seems to be aggravated by these factors and contribute to an increased fear of misdiagnosis and inappropriate clinical management.

GPs openly express concerns that digital virtual care could delay diagnosis and treatment. Respondents reported that these delays could be caused by patients' reluctance to use virtual care, lack of access and digital skills to use these tools, and communication challenges (i.e., to effectively convey non-verbal clinical information).

GPs also emphasised some potential threats to the efficiency of care delivery. While some patients expressed hesitancy to use virtual care, others were overly enthusiastic about having a more direct line of communication with their healthcare provider, often resulting in misuse and overuse. This increase in patient demand translated into an increase in GPs' workload. Finally, a few quotes highlighted that virtual care may not be an effective solution for certain types of consultations.

**Healthcare providers.** The lack of formal training, guidance, and inadequate technical support were notable challenges. During the abrupt shift to virtual care, GPs reportedly had to set up, learn how to use, and troubleshoot new systems. These efforts were seldom well-organised or coordinated, often resulting in considerable transitional difficulties. Another major drawback was the higher workload and burnout. Some participants reported that they found virtual consultations to be more taxing and time-consuming to perform, which was often burdened with increased administrative tasks. As described above, the frequency and ease with which GPs could be contacted by patients often blurred the boundaries between their work

Textbox 1. Main benefits: thematic analysis of the participants' narratives (Table is author's original work)

**THEME 1: BENEFITS FOR QUALITY OF CARE**

*Subtheme 1.1. Improved safety: Reduced risk of COVID-19 transmission*

- *"[Digital technologies] allowed decreasing the exposure to infection to both health providers as well as patients" (ID 312, Colombia)*

*Subtheme 1.2. Improved effectiveness: (access and continuity of care) and equity*

- *"Able to rapidly adapt to the new reality of COVID-19. We closed the front door but could still keep providing patient care." (ID 924, Ireland)*

- *"Continuing care with patients who otherwise might not have attended the practice because of concern about infection." (ID 1570, Australia)*

- *"the possibility of taking care of the patients even without seeing them. . . especially [ensuring the] patient did not feel abandoned." (ID 215, Italy)*

- *"[Digital virtual care] allowed us to remain accessible and in contact with our patients (ID253, Spain)*

- *"Continuity of care for patients with chronic diseases; psychological support or therapy for patients with mental health conditions remotely." (ID 347, France)*

- *"Facilitated the follow-up of diagnosed or suspected patients, favouring that isolation does not imply loss of care and detection of complications. Patients and their families or caregivers were supported" (ID387, Spain)*

- *"Able to continue to care for patients, especially those with acute needs or fragile chronic needs." (ID 566, USA)*

- *"[we were] able to provide care for people who have various barriers to accessing in-person care (frail elderly; people who cannot get time off work; people with physical disabilities)" (ID 1446, Canada)*

- *"Helpful for patients living far from the practice, with infectious diseases, disabilities. Helpful for follow up visits." (ID 827, Poland)*

- *"The ability to provide care in remote communities that otherwise have to family physician and would traditionally resort to presenting to the ER" (ID 1336, Canada)*

*Subtheme 1.3. Improved efficiency: triage and practice management*

- *"Ability to triage and bring in patients who need F2F: (ID 408, UK)*

- *"Allow qualified triage by doctors, giving access to face-to-face to the patients that do need it (ID 126, Portugal)*

- *"By doing a prior telephone screening, face-to-face assistance and follow-up could be better prioritized. Before, almost all patients requested a face-to-face appointment as soon as possible without any prioritization or filtering criteria. That was a great improvement." ((ID401, Spain)*

- *"Help filter patients and problems, so that only cases that are critical and need a prompt resolution make it to the healthcare centres" (ID 274, Chile)*

- *"Optimise health resources, especially human resources"* (ID 870, Chile)

- *'Optimisation of resources to answer the healthcare needs of the population'* (ID 327, Portugal)

- *'We have learned that many issues can be taken care of without a face-to-face consultation. This seems to be efficient for both the doctors and the patients'* (ID 10, Finland)

### Subtheme 1.4. Improved timeliness

- *"We had faster access to consultation with colleagues in hospitals for chronic patients, via e-referral without the colleague seeing the patient live."* (ID 1075, Croatia)

- *"Shorter waiting times to receive care"* (ID 149, Colombia)

- *Meetings with others are efficient, timely and seem easier. No time wasted with travel etc* (ID 833, UK&NI)

- *"Possibility of accessing consultations in a more expedited way"* (ID 436, Chile)

- *Ability to deliver care in a timely fashion. I was pleasantly surprised that the majority of visits were easily done on the telephone or via videoconference without compromising patient safety or satisfaction."* (ID 409, Canada)

### Subtheme 1.5. Improved patient-centredness: convenience, communication, and patient empowerment

- *"The biggest benefits have been increasing access to care and for allowing visits to be able to more seamlessly fit into patients' lives."* (ID 1282, USA)

- *"Reduce patient discomfort for access to care"* (ID 246, Italy)

- *"[Digital technologies] strengthened communication with patients to offer help and support"* (ID 1467, Colombia)

- *"With technological support [..] the relationship is integrated and modified, often enriched; this actually goes against the clichés of depersonalisation of relationships very often reported by professionals of my generation"* (ID 223, Italy)

- *"[Digital virtual care] allowed us to (. . .) delegate greater responsibilities in self-management of minor health problems* (ID253, Spain)

- *"(. . .) allowed us to remain accessible and in contact with our patients, and delegate greater responsibilities in self-management of minor health problems* (ID253, Spain)

- *"Educating patients, self-managing minor health issues and hopefully better communication"* (ID 1085, Croatia)

### THEME 2: BENEFIT FOR HEALTH CARE PROFESSIONALS

### Subtheme 2.1. Flexibility in location of work

- *'Possibility to work even if I was in quarantine'* (ID 124, France)

- *'Gave me the ability to work from home while I suffered from mild COVID.'* (ID 1516, Sweden)

- *"The possibility of working remotely allowed me to preventing burnout and conciliating family and professional demands"* (ID 47, Portugal)

- *"Telephone consultations normalized and kept people out of office who needed to be home—allowed clinicians to be away from office when sick and still provide care" (ID 1252, Canada)*

### Subtheme 2.2. More control over schedule

- *"Better management of appointments by the physician" (ID 55, Portugal)*

- *"More control for me over my schedule" (ID 110, Israel)*

### THEME 3: BENEFITS FOR HEALTH CARE SYSTEMS

### Subtheme 3.1. Hastening the digital transformation

- *"Working through video consultations was a very positive experience that I never would have tried without the pandemic.' (ID 809, Sweden)*

- *"[It was an] eye opening experience for both doctors and patients, [showing] that tele-medicine is helpful and can be safely used.' (ID 827, Poland)*

- *"It was a chance to experience the digital consultation and to see that primary health care can use more advanced digital solutions for taking care of patients. (ID 951, Turkey)*

- *"Learning that one can treat patients in many cases by their symptoms only without a frontal clinical exam" (34, Israel)*

- *"I realised that a lot of things can be done without face-to-face contact which leaves me more time for patients who need to be examined" (Slovenia, ID 838)*

- *"Realize that clinical and non-clinical activities can be performed at a distance saving time, transportation and energy, which could be maintained after the pandemic" (ID 193, Chile)*

- *"Active and transformative learning that a significant proportion of health concerns can be managed remotely (ID 40, Portugal)*

- *"It increased the digital skills of doctors and patients" (295, Italy)*

- *"Upgrade of electronic connection/data transfer between health fond, computer/program provider and family doctor" (ID 1109, Croatia)*

### Subtheme 3.2. Changing legal and regulatory frameworks

- *"We started online consultation in primary care which was prohibited in Poland before" (ID 556, Poland)*

- *"In Brazil, the biggest benefit was a political one, since only now teleconsultations were made legal in the country." (ID 79, Brazil)*

- *"I actually got paid for some of the stuff we normally have to do in lunch break or after hours for free.". (ID 1560, Australia)*

- *[Receiving] payments for such services" (ID 587, USA)*

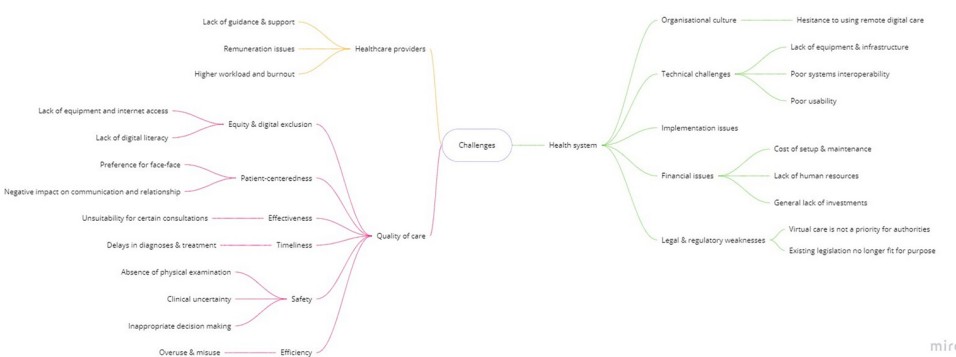

**Fig 2. Conceptual map of main challenges identified.**

and personal lives, further contributing to an increased workload. The increase in the number of patients observed daily has also resulted in less engagement with individual patients and decreased overall work satisfaction. GPs also reported inadequate remuneration as a considerable barrier to the continued use of virtual care. Whilst payment structures differed across countries, our respondents expressed that payment for remote consultation which is commensurate to in-person consultations, is critical to incentivising greater acceptance and mainstream use. As many of the remote consultation payment schemes were introduced during the initial months of COVID-19 pandemic as emergency measures, some GPs were doubtful that these efforts would be sustained over time.

**Health systems.**   Participants mentioned organisational culture as an impediment against their sudden widespread use. Overcoming substantial institutional inertia, the lack of pre-existing teleworking culture, the urgency to shift from familiar systems to completely new ones and challenging the prevailing mentality of '*this was always how things were done*', were commonly reported organisational barriers. Participants also reported technical issues with the digital systems used, including inadequate equipment and infrastructure, and poor systems interoperability. In particular, the lack of interoperability between several digital systems, such as appointment schedulers, electronic health records, electronic prescriptions, and epidemiological surveillance systems, was perceived as a major challenge. GPs described implementation issues related to finances, including a general lack of investment and, more specifically, lack of funding allocated to human resources and supporting the costs of systems set-up and maintenance. Respondents noted that virtual care delivery did not appear to be in the list of priorities of many healthcare systems or policymakers. Finally, GPs identified a range of legal and regulatory weaknesses, highlighting that existing legislation is no longer fit for purpose, nor meets the evolving needs resultant from the pandemic.

## Discussion

### Principal results

Benefits were identified in accordance with the six domains of quality of care. These included a reduction in exposure risks for COVID-19 transmission, ensuring access and continuity of care to those who need it, including those who had previously limited access to face-to-face consultations, enabling remote triage, in addition to improved patient convenience, communication, and empowerment. Benefits for healthcare providers included a greater work flexibility and more control over their schedules. Benefits for health systems included hastening the

Textbox 2. Main challenges: thematic analysis of the participants' narratives (Table is author's original work)

**THEME 1: CHALLENGES FOR QUALITY OF CARE**

*Subtheme 1.1. Challenges for patient-centredness*

- "Patient preference for face to face due to issues such as poor hearing on devices and patients not having ability to use technology." (ID 1561, Australia)

- "Patients and doctors prefer face to face, many patients pay in Ireland, online isn't acceptable [to them]" (ID 1019, Ireland)

- "Patients are accustomed to meet in GP clinics to talk with one another, especially older patients, lonely, they are waiting for meetings with doctors frequently too. Some have problems with electronic devices." (ID 556, Poland)

- "Lack of human contact, facial and body language. Difficult to show emotions and compassion. I used to close my eyes and imagine the patient on the other side of my table. I expend more time searching the suitable words to overcome the lack of body language" (ID 62, Portugal)

- "Difficulty to build rapport with patients who you don't already know—less trust of HCP" (ID 1033, UK&NI)

- "Harder for people who are socially isolated, coming to the clinic was an important human interaction, form of meeting other patients, the extended team etc." (ID 630, Canada)

*Subtheme 1.2. Challenges for equity, and digital exclusion*

- "Some elderly (the ones with the biggest need) have problems using a telephone, let alone a computer..." (ID 1518, Sweden)

- "Unfortunately, a lot of elderly patients (...) do not know how to use digital technology (email, messenger, etc) and in my opinion were deprived during the pandemic." (ID 1075, Croatia)

- "*[an unintended consequence was]* Neglecting the less technology-oriented patients, which are also the most vulnerable ones—the elderly, the underprivileged, the immigrants etc, and in fact giving preference to the younger healthier patients, thereby deepening health inequality" (ID 254, Israel)

*Subtheme 1.3. Challenges for safety*

- "[It is] much harder to make decisions on the phone without examining the patient. Often patients refused to come in for appointments and didn't understand the lack of ability to assess a certain presenting complaint over the phone (e.g., abdominal pain)." (ID 1598, Australia)

- "Not all necessary information concerning a patient's condition can be transmitted digitally/by phone and this causes a risk of not noticing a critical symptom/change in a patient's condition" (ID 8, Finland)

- "I found that I made more mistakes when making the initial diagnosis over email / our app / phone. If there is only a one-way written communication, then it is easier to get

seduced by the patient's view and not valorise objective facts, as we would in a face-to-face consultation." (ID 1320, Croatia)

### Subtheme 1.4. Challenges for timeliness

- "[A potential challenge was] not detecting patients with for instance atrial flutter because not seeing and examining them. More patients with symptoms of serious diseases are reluctant to seek care. Some are probably missed and come when the disease has progressed. I tried to be aware of this and arrange secure ways to investigate the patients." (ID 317, Sweden)

### Subtheme 1.5. Challenges for effectiveness

- "Telephone consultations instead of face-to-face are less effective in many cases. It works fine for minor problems" (ID 440, Sweden)

- "Harder to check up on chronic diseases patients (prescribing medications without physical examination), easier to miss potentially dangerous symptoms, harder to counsel patients in psychological distress (video calls are not common)" (ID 499, Poland)

- "Online tools that don't work as well as they should (for example don't provide all the necessary information for a professional or lead to another face-to-face consultation anyway)" (ID 18, Finland)

### Subtheme 1.6. Challenges for efficiency

- "Certain patients have been calling in much more frequently for minor issues that they would normally not have contacted their primary care provider for." (ID 409, Canada)

- "Make people understand what the priorities are and the appropriate times. I have kept the phone on for emergencies and have sometimes been contacted for irrelevant things at inconvenient times." (ID 169, Italy)

- "Too much access for system abusive patients" (ID 42, Portugal)

### THEME 2: CHALLENGES FOR HEALTHCARE PROFESSIONALS

### Subtheme 2.1. Lack of guidance and support

- "[One challenge was] the lack of previous training neither during undergraduate nor [during] postgraduate training. I have taken some online training. Also, the challenge was the lack of guidelines for primary care doctors." (ID 158, Poland)

### Subtheme 2.2. Higher workload and burnout

- "It takes more time to do a remote consultation. It requires preparing the call, reviewing the clinic history and having a checklist according to the patient's category. If a face-to-face consultation takes 20 min, a remote consultation requires 40–50 min average" (ID 244, Colombia)

- "I felt that my resources were drained by the phone communication. I am less concentrated, more nervous and less compassionate about patients" (ID 286, Israel)

- "Found myself working longer hours as working from home made it much harder to set boundaries around when I stopped work. (ID 1590, Australia)

- "I would contend that the majority of GPs would prefer to consult with their patients' in the real world more often than in the virtual world and that, for example, return to pre-COVID-19 levels of telephone or video consultations, if accompanied by a commensurate increase in in-person consultation, would be a positive development." (ID 910, Ireland)

### Subtheme 2.3. Remuneration issues

- "Prior to COVID-19, video and telephone consultations [were] not remunerated by Medicare. Technology was available, but was paid by the user/patient, prior to COVID-19. The telehealth subsidy in Australia is scheduled to end by March 2021, so we will return to dark ages again because patients might not want to pay for healthcare—especially telephone or video consultations, which are seen as low value by patients preferring face to face and hands-on medical care." (ID 1603, Australia)

- "If government withdraws fees for phone and video consultations then I would be less inclined to use them and more inclined to bring patients into office (sometimes needlessly)" (ID 1245, Canada)

### THEME 3: CHALLENGES FOR HEALTH CARE SYSTEMS

### Subtheme 3.1. Organisational culture

- "Tradition: "we have to return to what we did before. . . it worked. . . "" (ID 178, Chile)

- "People don't like changes, especially if they have to do something differently, or additionally" (ID 1096, Croatia)

- "People [will] forget easily how digital ways of working facilitated work during the pandemic and [will] easily return to the familiar normality which they knew before the pandemic" (ID 10, Finland)

### Subtheme 3.2. Technical challenges

- "There is no resources (computers, telephones) for all health workers, therefore not everyone can work through digital techs and personal resources need to be used such as personal phones" (ID 274, Chile)

- "EHR and network problems—overloaded, not prepared for massive digital use of systems, working very slowly. Still waiting for the solution (new server, better internet connection, etc)" (ID 55, Portugal)

- "Lack of efficient digital equipment and software able to communicate with each other, because up to now it means opening many windows over and over again" (ID 199, Chile)

- "No integration between systems and some of them do the same thing instead of complementing each other" (ID7, Portugal)

### Subtheme 3.3. Implementation issues

- "The technologies were fine, but my health system did a terrible job implementing the changes. They were slow to find video services. They changed the type of video service three times." (ID 566, USA)

### Subtheme 3.4. Financial issues

- "They furloughed so many staff personnel that physicians were expected to do digital appts with no staff support. There was no staff to answer phones except two days a week for two weeks. It was a mess." (ID 566, USA)

- "Public organizations inability to see investments—only costs. Conservative culture within the organization." (ID 1414, Sweden)

- "Lack of financial investment from governments in such technologies; Patients emotional need to get in touch with medical/other health care workers staff; Specially in Brazil, it is possible that our Medical Council pressure up against telemedicine consultations because it is an issue that before the COVID19 pandemic situation they have never been sympathetic with." (ID 65, Brazil)

- "Short-sightedness of managers, Unequal distribution of resources, inbred inertia of large systems which resist change" (ID 11, Israel)

*Subtheme 3.5. Legal and regulatory weaknesses*

- "Current legislation limits teleconsultations" (ID 860, Chile)

- "Regulatory barriers across state and geographic lines—overly strict (. . .) regulations —prevent right care in right venue direction" (ID 1378, USA)

- "In my opinion, the main problem lies in the fact that, according to the Italian code of medical ethics, the medical examination can only take place in presence. This limit has significant repercussions on the lawfulness of the prescription and certification carried out during a video consultation." (ID 175, Italy)

digital transformation through increasing awareness, trust, adoption, skills, and technical capacity, as well as driving changes in legal and regulatory frameworks.

Likewise, significant challenges have also been highlighted across the six domains of quality of care. These included patients' preference for face-to-face care, the potential negative impact on communication, and the lack of equipment, internet access and digital skills of some patient groups. In addition, clinical uncertainty and potentially inappropriate decision making resulting in delays in diagnosis and treatment, unsuitability for certain consultations, as well as overuse and misuse of healthcare resources, were also mentioned. Challenges specific to healthcare providers included the lack of guidance and support, higher workload, and remuneration issues. From the health systems' perspective, the long-established organisational culture, technological difficulties, implementation and financial issues, and inadequate accompanying supportive policies and regulatory legislation, were also challenges described by participants.

## Strengths & limitations

This is the first international study to explore GPs' perceptions on the main benefits and challenges of using virtual consultations in primary care. Participants took part from 20 countries worldwide, with diverse health care systems and levels of healthcare spending. The sample size was large, with participants varying in age, clinical experience, and type of primary care setting (urban, rural, or mixed). This study employed a methodologically rigorous approach,

leveraging qualitative methods to capture rich, descriptive data on individual perceptions, attitudes, and behaviours [35,36], performed according to the Consolidated Criteria for Reporting Qualitative Studies criteria [37]. Finally, the main benefits and challenges were mapped against a widely recognised framework for Quality of Care [10], whenever possible. Finally, a set of recommendations was developed based on the main findings, to support providers and healthcare organisations translate the lessons learned into practice improvements.

The results must be interpreted considering some limitations. Our findings are impacted by common limitations of survey research, including self-reported answers and self-selection sampling methods. The predominance of GPs working in predominantly urban settings (62%) might be a consequence of the sampling method or represent actual geographic variations on delivery of virtual care. While our results do not allow to draw specific conclusion on this matter, future research should aim to clarify geographic variations, within and between countries, in what concerns the availability and use of virtual primary care. The predominance of the urban setting and the fact that urban GPs' likely work as part of multidisciplinary teams may have influenced their responses. It is worth noting that approximately 40% of our responses were still derived from GPs in rural or mixed settings, and thus should still allow for differing views to be captured.

Only GPs were included in this study; future research should focus on the inclusion of other healthcare professionals and patients. The themes identified as part of this analysis and the subsequent recommendations derived from them, may not be equally relevant for each individual country given the diverse forms of virtual care used and the differing COVID-19 induced healthcare demands of the respective national health systems. Further qualitative content analysis could provide novel insights on their relative importance for individual countries, and specific groups of GPs. It is also important to note that this study evaluates qualitatively GPs perceptions on the impact of remote care; future research using quantitative approaches to objectively evaluate potential changes observed is critical to validate these findings.

Finally, virtual care is a broad concept and future research must explore specific nuances of the various types of technology available (i.e., telephone, video, chat), both in what concerns perceived benefits and challenges of implementation, and patient preferences.

## Comparison with prior work

Remote primary care is widely recognised as an promising solution to ensuring both patient and provider safety by preventing direct physical contact, hence reducing morbidity and mortality during the COVID-19 pandemic [16,38,39]. However, important safety concerns also manifested, predominantly concerning diagnostic uncertainty. In this context, previous literature also report GPs' concerns about clinical risk [40] and the need to establish escalation protocols to support clinicians decide when a transition to urgent in-person follow-up care, or even to emergency services, is required [39].

Our results underline that remote digital tools may be an effective way of delivering primary care. In line with these findings, a Cochrane systematic review (2015) demonstrated the use of telemedicine strategies to be associated with increased access to care and improved clinical outcomes in single chronic diseases, particularly in type 2 diabetes [41]. However, the interventions were heterogenous and the external generalisability of these findings remain unclear. Future research will be needed to address questions such as for which patients, and for which conditions, do virtual care tools actually improve effectiveness.

Participants highlighted that virtual care, particularly through remote triage, can reduce the number of unnecessary visits and thus have a positive impact on efficiency (i.e., minimising waste, including from an economic perspective). Few telehealth evaluations have examined the

association between outcomes and costs of virtual care. While some reviews have found that virtual care can decrease the use of acute hospital services [42–44], there is less evidence in the primary care context [45]. On the other hand, our participants raised concerns about potential overuse and misuse by patients. In a recent study in Canada (2020) evaluating the uptake of a platform for virtual visits in primary care, Stamenova *et al.* observed that many virtual visits appeared to replace face-to-face visits, yet patients did not overwhelm physicians with requests [46].

Regarding timeliness of care, participants identified both potential advantages and disadvantages. Remote primary care has the potential to offer convenient access to a primary care provider without needing to take time-off work, arrange transportation, and spend time waiting for face-to-face visits. Participants were also concerned that barriers to the use of technology and difficulties inherent to a new mode of care delivery, could result in delays in diagnosis and treatment. There is sparse evidence on the subject. However, a recent study examined patient-initiated primary care visits in Kaiser Permanente Northern California (a system with over four million members) and concluded that, on average, telephone visits were scheduled 50% sooner than office visits [47]. These findings have profound implications, given that timeliness of care is associated with improved health outcomes [48].

Virtual care has been promised to reduce inequities in access to care for decades, particularly in rural and geographically remote areas [49–51]. In line with previous literature, our results demonstrate the ability of virtual care to overcome barriers for those who have physical limitations to attend a face-to-face meeting, but also highlight their potential to entrench existing inequities in access to care [52,53]. Published evidence shows that the transition to virtual primary care did not unfold in the same manner across communities [54]. Proactive efforts are therefore needed to identify and address both patient and provider-related digital barriers to avoid that the widespread implementation of virtual care in a manner which reinforces disparities in health access amongst already underserved and excluded groups [54]. Future research should also evaluate differences on the different types of solutions available in each country, whether these are free of charge or nor, and whether they are developed by private companies.

Equally, with regards to patient-centredness, a range of benefits and challenges have been identified. While participants consider that virtual care can improve convenience and patient empowerment, participants also acknowledge that it can have both negative and positive effects on communication. Another important challenge is patient preference for face-to-face visits. Preference theory suggests that patients will prefer a virtual consultation if they perceive its benefits as outweighing its burdens [55]. Multiple factors may influence patients' preference, including the situation of care (i.e., patient's perception of their clinical status, treatment requirements, and care pathway), the expectations of care, the demand of care (e.g., social situation, consequences of choice), the capacity to allocate resources (e.g., patient's ability to allocate financial, infrastructural, social and healthcare resources) [56] and patients' digital health literacy [57,58]. These factors may combine or compete and may be dynamic throughout the patient's journey.

The pandemic has a transformational impact in hastening the digital transformation, and in particular increasing awareness, trust, and adoption of virtual care [16,59,60]. Challenges such as the lack of support, burnout, and remuneration issues, are balanced by a few benefits (i.e., flexibility in work location and a better control over schedule). However, to fully embrace the benefits postulated (e.g., greater flexibility to work remotely from home), healthcare systems must explore and implement the enablers required. Potential enablers include single sign-on, sharing of records across systems, electronic generation of form, and the creation of workspaces and secure interfaces without the need for laptops and VPNs [61]. While the rapid

implementation of virtual consulting tools provides the ability to work more flexibly and from various locations, primary care leaders need to be supported and learn how to build effective teams via novel approaches [62].

At the health systems level, previous studies have indicated that implementation barriers depended on accreditation, payment systems, and insurance [63]. Prioritisation of financial investments into relevant infrastructure, greater emphasis on healthcare providers training, and updates to the corresponding legal and regulatory frameworks supporting their use, are equally required. Overcoming institutional inertia is likely to be more feasible post-COVID-19, given that clinical culture is expected to have evolved substantially after a year of daily use of virtual care delivery. As many of the existing guidance and policies on primary virtual care was drafted and implemented during the emergency phase, the experience attained, and evidence collected offers an opportunity to refine, optimise, and update the relevant accompanying legal and regulatory frameworks.

## Implications for policy and further research

Our findings highlight the complexity inherent to the implementation of virtual care solutions in primary care settings and underscore the need to adopt tailored strategies to address the challenges identified, as well as to enhance the potential benefits postulated.

Several factors and policies can contribute to successful virtual care implementation. These include recommended actions to address contextual considerations, technology infrastructure, awareness & experience, safety & risk management, strategic planning and supporting policies (Fig 3). As part of these recommendations, we emphasise the importance of continuously monitoring quality across these five areas, which must be collected both through patient and provider feedback, but also using data-driven approaches to systematically evaluate the impact on equity, effectiveness, safety, efficiency, and timeliness. Decisions to adopt virtual care services should be based on evidence of their impact, and be responsive to ongoing evaluation & monitoring processes.

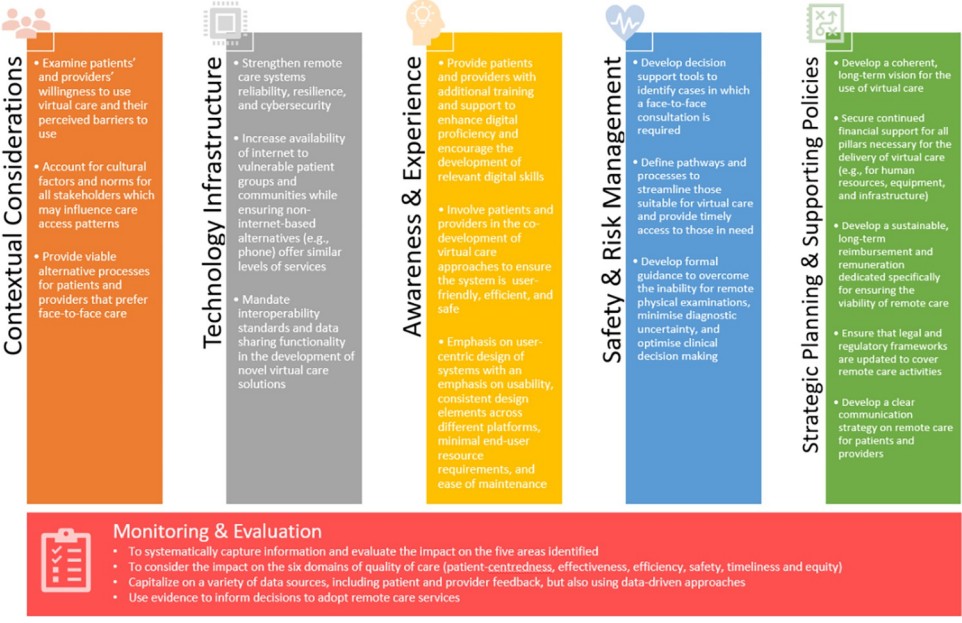

**Fig 3. Summary of recommendations for the implementation of virtual consultations in the primary care setting.**

It is important to note that the technological and regulatory landscape is dynamic, as are patients' preferences. Therefore, virtual care platforms must be secure and reliable, but also flexible enough to accommodate evolving regulatory, professional, and health-care organisations' requirements [64]–and processes must ensure that patients are presented with alternative options during their journey.

## Conclusions

At the frontlines of care delivery, GPs can provide valuable insights into the use of virtual care. Experience gained during the COVID-19 pandemic can be used to inform the stable adoption of virtual care solutions, the co-designing of processes and platforms that are technologically robust and supported by a strategic long-term plan. Such platforms should consider pre-existing health inequities and contextual considerations. Accompanying policies need to minimise digital exclusion, optimise patients' experience, and necessitate rigorously evaluations of virtual primary care both during and after the pandemic, and incorporate the lessons learned into legal and regulatory frameworks to support its long-term, sustainable use.

## Supporting information

**S1 Appendix. Questionnaire assessing GPs' perspectives on the adoption and experience of virtual care solutions during the COVID-19 outbreak.**
(PDF)

## Author Contributions

**Conceptualization:** Edmond Li, Ana Luisa Neves.

**Data curation:** Edmond Li, Ana Luisa Neves.

**Formal analysis:** Edmond Li, Ana Luisa Neves.

**Funding acquisition:** Ana Luisa Neves.

**Investigation:** Edmond Li, Ana Luisa Neves.

**Methodology:** Edmond Li, Ana Luisa Neves.

**Project administration:** Ana Luisa Neves.

**Supervision:** Ara Darzi, Ana Luisa Neves.

**Writing – original draft:** Edmond Li, Ana Luisa Neves.

**Writing – review & editing:** Edmond Li, Rosy Tsopra, Geronimo Jimenez, Alice Serafini, Gustavo Gusso, Heidrun Lingner, Maria Jose Fernandez, Greg Irving, Davorina Petek, Robert Hoffman, Vanja Lazic, Ensieh Memarian, Tuomas Koskela, Claire Collins, Sandra Milena Espitia, Ana Clavería, Katarzyna Nessler, Braden Gregory O'Neill, Kyle Hoedebecke, Mehmet Ungan, Liliana Laranjo, Saira Ghafur, Gianluca Fontana, Azeem Majeed, Josip Car, Ara Darzi, Ana Luisa Neves.

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
