## [Decision Letter · Decision Letter 0]

23 Feb 2022

PDIG-D-21-00123

General practitioners’ perceptions of using virtual primary care during the COVID-19 pandemic: An international cross-sectional survey study

PLOS Digital Health

Dear Dr. Li,

Thank you for submitting your manuscript to PLOS Digital Health. After careful consideration of the submission and the encouraging peer reviews we received, we feel that it has merit but does not fully meet PLOS Digital Health's publication criteria as it currently stands. Therefore, we invite you to submit a revised version of the manuscript that addresses the points raised during the review process.

We look forward to receiving your revised manuscript.

Kind regards,

Pauline Bakibinga, M.D, Ph.D

Guest Editor

PLOS Digital Health

Journal Requirements:

Additional Editor Comments (if provided):

Reviewers' comments:

Reviewer's Responses to Questions

**Comments to the Author**

1. Does this manuscript meet PLOS Digital Health’s publication criteria? Is the manuscript technically sound, and do the data support the conclusions? The manuscript must describe methodologically and ethically rigorous research with conclusions that are appropriately drawn based on the data presented.

Reviewer #1: Partly

Reviewer #2: Yes

Reviewer #3: Yes

2. Has the statistical analysis been performed appropriately and rigorously?

Reviewer #1: I don't know

Reviewer #2: I don't know

Reviewer #3: No

3. Have the authors made all data underlying the findings in their manuscript fully available (please refer to the Data Availability Statement at the start of the manuscript PDF file)?

Reviewer #1: No

Reviewer #2: Yes

Reviewer #3: No

4. Is the manuscript presented in an intelligible fashion and written in standard English?

Reviewer #1: Yes

Reviewer #2: Yes

Reviewer #3: Yes

5. Review Comments to the Author

Reviewer #1: Thank you for your submission and the opportunity to review this. This is a topic of interest to policymakers, administrators and of interest to people who develop either telehealth platforms or models of care based on telehealth. That this is multinational adds value over any prior work and I think is worth progressing to publication in PLoS Digital Health.

There are some challenges with the manuscript- partly this is related to the cross-disciplinary nature of the journal where, say, engineers, policymakers, social scientists and clinicians are interested in each other's work and need to be able to understand them as much as possible from the text. With regards to the criteria for publication:

- I think the methods section is a bit light- not so much in length but on the basis of reproducibility.

- The analysis sub-section of the methods in particular needs a clearer description of whether this is a previously used method or new, the existing references don't seem to be that helpful.

- If this is qualitative rather than largely quantitation, what is the epistemology and methodology?

- The survey seems to include a lot of structured fields, so presumably some quantitation has occurred, so this would be helpful to clarify.

- The results section is really interesting and I think this is valuable. I immediately went looking for a comparison with prior work and like that this is clearly headed. Was there anything you expected to find but didn't?

- I partly agree with your conclusion, but would like to see the connection with models of care more clearly outlined.

- Surprisingly there doesn't appear to be any sub-group analysis, in particular by country. In fairness, there may be issues with relative number and this may be a big enough piece of work that it could be a separate publication, but I do think there are some important comparative questions which you're in a unique position to explore. One of the factors that immediately came to mind for me being in Australia is that the funding mechanism for telehealth for GPs is Medicare, that has undergone a lot of changes through COVID and this quasi-regulatory mechanism considerably influences what is delivered and how (since it's how GPs get paid). Although you have a wide range of health systems included, most of which have universal healthcare, the means by which universal healthcare is implemented and the mechanism of health insurance is relevant.

- It would be helpful to include de-identified data as a supplementary file, if ethics allow for this.

Reviewer #2: Methodology:

There don’t appear to be hypotheses.

Who were the two independent researchers who reviewed transcripts? Were they GPs? Were they members of the inSIGHT group? It is difficult to assess how independent they were without this information.

It was unclear whether some/all GPs were seeing both patients inhouse and virtually? (hybrid)

There were some comments as to the types of care in which virtual visits were more effective, but it does not appear these were collected systematically.

The amount of time spent with patients for virtual and inhouse consultations was not obtained--limitation. NOTE: at particular phases of the pandemic, patients were reluctant to see their primary care physicians which likely freed up time for the practicing GP. Thus, it is possible the GP spent more time with patients for virtual care.

What about diagnoses that have no symptoms which become apparent from clinical exams, blood and/or urinalysis? These can certainly be more detectable from annual/routine physical exams which collect labs, assess weight gain, substance use, stress, etc.

If patients have a virtual consultation which necessitates an inhouse visit, does the patient get charged a second co-pay? This could promote health disparities.

The type of healthcare system may impact on differences in perception—it is noted but not discussed.

Discussion

Strengths & Limitations: The phrase “recommendations for the implementation of virtual consultations in the primary care setting” seems a bit premature without input from patients. Most if not all are suggested strategies to facilitate implementation of virtual consultations that need to be discussed and evaluated. 

Limitations: using virtual consultations, coloration of skin may not be true.

Limitations: The predominance of respondents from urban settings may be explained by where the country leaders had their practices, given that snowball sampling was used (i.e., urban, suburban, rural)—those data were not presented.

Limitations: GPs were not asked as to the distribution of social determinants of health among patients in their practice which may be related to the frequency and time spent with patients due to prevalence of comorbidities.

Limitations: Types of virtual patient interfaces were not requested. These can include telephone, email, still pictures/videos, facetime, zoom or other platforms.

Line 318: Limitations: note regarding “future research should aim to clarify … .” Given the topics specified, it would be appropriate to plan and conduct a (primarily) quantitative survey.

Inferences:

Line 344: Statement to introduce the paragraph is not totally correct—it is incomplete: “Our results underline that remote digital tools may be an effective way of delivering primary care.” The statement should reflect the type of patient in whom remote care may be beneficial relative to the risks.

Line 325 and beyond: Future research should include more standardized methods of virtual care delivery, that could better interface with public health reporting. Standardization is alluded to in Figure 3—it facilitates transfer of data and it’s use for research and reporting.

Line 421: Certainly, monitoring quality is a priority. However, it should be noted that COVID-19 impacts quality outcomes so trends can be deceiving unless adjusted.

Grammar/Spelling: 

Subtheme 1.5. “(...) allowed us to remain accessible and in contact with our patients, and delegate greater responsibilities in self-management of minor health problems (ID253, Spain) the second clause is identical to that which preceded it—could be deleted in the second quote

GPs openly expressing concerns that digital virtual care could delay diagnosis and treatment. ??Should be ‘openly expressed …’

Discussion: “performed according to the Consolidated Criteria 311 for Reporting Qualitative Studies criteria” should be “reported’ according to …

There are two sentences in the Strengths and Limitations section of the Discussion that begin with ‘Finally”.

Discussion (Strengths & Limitations) last sentence of paragraph, “as set of recommendations was developed..” There should be a reference to the figure 3.

Line 320: “the availability and use or virtual primary care.” Should be “… use of virtual …”.

Line 406: healthcare providers’ training,

Line 341: “need to establish escalation protocols to support clinicians decide when a” should be “… support clinicians’ decisions …”

Reviewer #3: 270 Participants mentioned organisational culture as an impediment against using virtual care. 

273 Participants also reported technical 

> Do you mean all participants in every country mentioned organisational culture as an impediment against using virtual care?

> Participants also reported technical issues with the digital systems used, including inadequate equipment and infrastructure, and poor systems, which participants (male, female, country)? Or overall percentage from the selected participants? 

274 issues with the digital systems used, including inadequate equipment and infrastructure, and poor systems 

275 interoperability.

> It is not clear which countries are related to the statement in lines 274-275. Can I conclude that all countries included in this study have same issues (inadequate equipment and infrastructure, and poor systems interoperability) ?

281 GPs identified a range of legal and regulatory weaknesses, highlighting that existing legislation is no longer 

282 fit for purpose, nor meets the evolving needs resultant from the pandemic

> Which GPs. How these GPs are distributed based on your selected characteristics. What is the percentage of countries that have legal and regulatory weaknesses ? do you mean that all countries or X% have legal and regulatory weaknesses?

286 Benefits for quality of care were identified for the six dimensions of quality of care: safety, effectiveness and equity, efficiency, improved timeliness and patient-centredness.

295 Important challenges have also been noted for six domain of quality of care: patient-centredness, equity, safety, timeliness, effectiveness, and efficiency.

> It is not clear why authors use six dimensions of quality of care and six domains of quality of care? As six dimensions of quality of care and six domains of quality of care contain almost words.

> Do you mean six domain or six domains (plural)?

325 Only GPs were included in this study; future research should focus on the inclusion of other healthcare 

326 professionals, and especially patients

> What are other health care professionals. I understood that “Especially patients” are one of other health care professionals or health care professionals that are patients. Can you please rewrite it to avoid this confusion? "Especially" term creates confusion. If the term "especially" will be left out, then it will create no confusion. 

313 recommendations was developed based on the main findings, to support providers and healthcare 

315 The results must be interpreted considering some limitations. Our findings are impacted by common 

332 evaluate potential changes observed is critical to validate these findings.

414 Our findings highlight the complexity inherent to the …., 

> Authors used findings in lines 313, 315, 332 and 414 without clearly define their findings in their paper. It is not easily to point out the findings of which the authors talk about in their paper. 

> Caption in Fig 2: "Conceptual map of main challenges identified" is not clear. The figure 2 is not clear.

Overall, the above questions are related to minor issues. The paper is of high quality that covers interesting topic in digital health. I salute the authors for their excellent work.

6. PLOS authors have the option to publish the peer review history of their article (what does this mean?). If published, this will include your full peer review and any attached files.

**Do you want your identity to be public for this peer review?** For information about this choice, including consent withdrawal, please see our Privacy Policy.

Reviewer #1: Yes: Dylan A Mordaunt

Reviewer #2: Yes: Katherine Freeman

Reviewer #3: No

---

## [Editor Report · Decision Letter 1]

28 Mar 2022

General practitioners’ perceptions of using virtual primary care during the COVID-19 pandemic: An international cross-sectional survey study

PDIG-D-21-00123R1

Dear Dr. Li,

We are pleased to inform you that your manuscript 'General practitioners’ perceptions of using virtual primary care during the COVID-19 pandemic: An international cross-sectional survey study' has been provisionally accepted for publication in PLOS Digital Health.

Best regards,

Martin C Were

Section Editor

PLOS Digital Health

Reviewer Comments (if any, and for reference):

The authors have adequately addressed all comments by reviewers.